# Critical Thinking in Ethical and Neutral Settings in Gifted Children and Non-Gifted Children

**DOI:** 10.3390/children10010074

**Published:** 2022-12-30

**Authors:** Rosa Angela Fabio, Alessandra Croce, Chiara Calabrese

**Affiliations:** 1Department of Economy, University of Messina, via dei Verdi, 75, 98122 Messina, Italy; 2Department of Clinical and Experimental Medicine, University of Messina, via Bivona, 98122 Messina, Italy

**Keywords:** critical thinking, giftedness, ethical and neutral, phases of critical thinking

## Abstract

The present study examined the performance on five phases of critical thinking in gifted and nongifted children in two settings: ethical and neutral. Ninety-one children, 32 gifted (8–10 years old), 32 normally developing children matched for chronological age (8–10 years old) and 27 normally developing children matched for mental age (12–13 years old) completed critical thinking tasks. The findings confirmed that intellectually gifted children had higher critical thinking capacity than typically developing children. The results reveal that the basic factor determining best performances in critical thinking is mental age and not chronological age. However, critical thinking ability was the same in ethical and neutral settings. Analysis of the phases of critical thinking show that the first and the third phase, clarification and evaluation, specifically differentiates gifted from nongifted children. These phases refer to the ability to understand the type of problem rapidly and to assess the credibility of statements and to assess the logical strength of the actual or intended inferential relationships among statements, descriptions, questions or other forms of representation.

## 1. Introduction

To think is human, everyone thinks, however, not everyone thinks well [1]. Indeed, human thinking is often distorted, biased and full of prejudices [2]. Some authors suggest that critical thinking (CT) is much more than good thinking; instead, it is thinking with logic or reasoning [3,4,5]. CT has been defined in many ways. It is targeted on self-regulatory processes and consists of interpretation, analysis, inference and interactive evaluation phases [6,7,8,9]. The CT process guides individuals to develop focused and reasoned judgments based on their beliefs [6,8]. Sternberg [10] defined it as the mental processes, strategies, and representations people use to solve problems, make decisions, and learn new concepts, while Willingham [11] defines it as the act of seeing both sides of an issue, being open to new evidence that may disconfirm your ideas, reasoning dispassionately, demanding that claims be backed by evidence, deducing conclusions from available facts, and solving problems. Although there are many definitions of CT, some phases characterize it constantly: clarification, analysis, evaluation, inference, self-monitoring [12,13]. Ennis [1] also considers self-assessment important, although he considers it a secondary element compared to the previous ones.

The skill of CT can help children and adults to overcome prejudices and errors of thinking; for this reason, CT is important in early childhood. For example, long before children begin school, they take in a vast amount of information from people and their surroundings; if children are not supported in analyzing this information, they are in danger of being misled [14]. With the vast amount of information available now compared to 30 years ago, it is even more important for children to learn to filter this information through CT [15].

### 1.1. Critical Thinking in Intellectually Average Children

CT in children is useful in order to participate in social discourse, as children have to learn to assess whether statements of disagreeing interlocutors are supported by compelling reasons, regardless of the reputation of the speaker [16]. Some studies propose that children from as early as 3 years demonstrate early CT [17,18]. Starting from an early age, children acquire much information from the statements they hear [19]. It has been shown that at the age of 2 they have the ability to distinguish strong from weak reasons presented in interpersonal discourse [20]. At the age of 4, it seems that children trust informants who have proven reliable in the past [21,22,23,24,25,26,27,28]. Around preschool age, children demonstrate sensitivity to the quality of reasons provided by their parents [29]. In addition, children have shown that they can respond to good reasons when engaging in cooperative decision-making with a peer [30,31].

### 1.2. Critical Thinking in Intellectually Gifted Children

Intellectually gifted students have specific characteristics that could predict CT skills. They generally process information faster than average ability peers on both simple and complex tasks [32,33]. Furthermore, gifted students are generally more thorough problem solvers than average ability peers [34,35]. Gifted students have also demonstrated a wider variety of strategies during problem solving than age peers [36,37]. There is also evidence that gifted students employ more metacognitive strategies during learning than their nongifted peers [38,39], and gifted students are generally better at assessing their abilities for a learning task than their nongifted peers [40,41]. Then, gifted students are generally able to keep their attention on a problem or task in ways their nongifted peers cannot [41,42,43,44,45,46].

Furthermore, a central feature of gifted children is their moral sensitivity. The relationship between talent and moral development is complex [47]. In reality, it is not necessary to be gifted to be moral, and the gifted are capable of incredibly destructive and immoral behavior. However, many studies have observed that gifted children express moral concerns at a younger age and more intensely than their peers, and some theorists suggest that moral sensitivity increases with intelligence [47,48,49]. Gifted children have been proven to promise high moral development in adult life. Their ethical sensitivity derives from their heightened cognitive awareness, acute sense of justice, emotional sensitivity, empathy, intuition, observation skills, knowledge of consequences, questioning the morality of culture and the ability to imagine alternatives [47,50,51,52,53,54].

With reference to the moral processes a study of Karnes & Brown (1980) indicates that the level of moral judgement in gifted students is positively related to the level of intellectual functioning. Also, Sanders et al. (1995) studied whether intelligence affects moral development as assessed across a range of different moral transgressions, and they saw that gifted individuals earned significantly higher moral reasoning scores than did their average-ability peer; they also scored higher than college freshmen who were 4 to 5 years older [55,56]. Furthermore, with reference to moral processes, Derriberry & Barger (2008) evaluated reaction time and attributional complexity as contributing factors to the relatively high moral judgment of gifted youth compared to college students. The subjects responded to a computerized measure of the development of moral judgment, which also indexed reaction time. The study found that the development of moral judgment of gifted young people is markedly different from the sample of college students, gifted young people tended to process moral information more quickly. Also, gifted young people prefer to process certain types of information more complexly than college students and have the ability to do so more quickly than college students. The gifted youngster reacted significantly faster to elements of self-interest and compliance with regulatory patterns [57].

It is also interesting to consider that the study of Beißert & Hasselhorn, (2016) recently found an absence of correlation between moral development and nonverbal intelligence in children aged between 6 and 8 years. Their study had the aim to verify whether morality and intelligence were related in younger children, as morality and intelligence had already been shown to be related in adolescents and adults. The results shown that moral developmental status seems to be independent from children’s general intelligence [58].

A study by Gilmanshina et al. [59] shows that gifted children are distinguished by peculiarity of thinking. These children are characterized by greater speed of imagination processes and are not prone to stereotyped decisions. Indeed, their thought process is characterized by a thorough analysis of the details and they tend to consider more options: as CT increases so does their creativity [59]. However, in literature, little research has been done on the development of CT in gifted children. The research mentioned above, [16,18,20,27,28,29], focused on the study of the development of CT in intellectually average children, showing that CT develops in infancy, but there are few studies on intellectually gifted children [5,47,51,60,61,62,63]. For example, it is not known if there is a difference in the development of CT in gifted children, and for this reason our research aims to make an innovative contribution to literature, investigating possible differences in the CT of intellectually gifted children compared to intellectually non-gifted children.

The theoretical assumption of this work is based on the contributions of the various previously mentioned research [1,10,11,13,21]. As seen, prior studies have shown that CT capacity is very high in intellectually gifted individuals, but no studies have explored the precise phase or process that differentiates gifted from non-gifted children. In particular, we aim to measure the performances of gifted children in the five phases of CT: (a) clarification, that is, the ability to focus on a problem and give it meaning; (b) analysis, the ability to identify the relationship between the parts of a problem and to distinguish what is relevant from what is not; (c) evaluation, the ability to ascertain the value of the sources to verify reliability, agreement, and credibility of information; (d) inference, the ability to come to a coherent and reasonable conclusion based on the information analyzed; (e) self-monitoring, the ability to know how to monitor cognitive procedures throughout the process, starting from self-observation up to self-correction.

Moreover, as it emerged from some studies [48,49,50,52] that gifted children express moral concerns at a younger age and more intensely than their peers, in this study it was decided to include an ethical component to investigate if there are differences in CT processes between intellectually gifted children and intellectually non-gifted children in a CT task with an ethical problem. This study focuses on an analysis of the behavioral patterns of the intellectually gifted children during the decision-making process. The general aim of this research was to verify the existence of any differences in the ability to use CT by intellectually gifted children compared to intellectually average children.

In particular the research objectives were:

In relation to the results of previous literature [38,44,45,46] to investigate whether gifted children show a higher level of performance in CT abilities than non-gifted children;To examine whether in some phases of CT (clarification, analysis, evaluation, inference, self-monitoring)***,*** and possibly in which ones, the differences described above are particularly significant.To understand if significant differences emerge in the ability to use CT, between gifted and nongifted children in two different types of settings one ethically neutral and the other with ethical implications [63].

## 2. Materials and Methods

### 2.1. Experimental Design

The experimental design is a split-split-plot design with three factors: one between subjects and two within subjects.

The factor between subjects is groups (intellectually gifted children, normally developing children matched for chronological age, and normally developing children matched for mental age). Instead, the variables within subjects are content (neutral vs ethical) and phases of CT.

### 2.2. Participants

The initial sample of subjects was made up of 932 children attending a primary school, 530 females and 402 males, from three primary schools in Milan, attending from the first to the fifth grade. They had from middle to high socioeconomic status (SES). The average household size was 3.20 (±1.2). The majority of individuals in the sample were Italian (*n* = 848; 91%), the 9% were Asian and African. Only the 5% of the sample were immigrants.

Raven Intelligence Measurement Test—Progressive Matrices [60] was administered to all 932 children, and on the basis of the IQ scores obtained (≤130), the experimental group was formed with 32 children with a mean IQ = 137.6 (±6.2). Following this first screening for giftedness, we obtained a non-gifted sample of 900 children. From this remaining group we extrapolated firstly 32 normally developing children matched for chronological age and gender; each child of this second group was matched one-by-one with each gifted child based on the same chronological age and the same gender; from the same group of 900 non-gifted children we extrapolated secondly 27 normally developing children matched for mental age and gender; each child of this third group was matched one-by-one with each gifted child based on the same mental age and the same gender (Table 1). The first chronologically age matched control group (mean CA = 9.12 (±0.98)), consisted of 32 intellectually nongifted children, with the same chronological age and the same gender of gifted children (expressed in years and months). Then, the second mental age matched control group consisted of 27 intellectually non-gifted children, matched by gender and mental age compared to the gifted group, but with a higher chronological age (mean CA = 11.8 (±1.02)).

The final sample consisted of 91 children divided into the three groups described above (experimental group of gifted children, control group for chronological age and control group for mental age).

Table 1 shows the demographic statistics of the sample of 91 subjects.

### 2.3. Measures

Raven’s Progressive Matrices Test was used to measure the IQ of the children [60]. The test is a non-verbal ability test typically used to assess abstract reasoning and general human intelligence and, since it is a non-verbal test, it is used to reduce cultural biases. This test is a progressive test as the items get harder and harder as the test progresses. The task is to determine the missing element in a pattern which is generally presented in the form of a matrix. The main component is the ability to reason, i.e., the reasoning that is done with respect to inclusion in a matrix. The test requires working memory, short-term memory, long-term memory, processing speed, executive functions, metacognition, controlled attention and expertise.

In order to evaluate the use of CT in the subjects belonging to the three identified groups, a test for CT was constructed, taking into consideration the five previously identified phases, which is the paradigm of this research. Two tracks have been created (Track 1 and Track 2). The first task was defined as “neutral” as it has no ethical implications. The problem lies in the choice of sport to be practiced by a child attending primary school. The second task also presents a possible real-life situation for a primary school child. Unlike the first passage, it includes ethical aspects such as respect for the rules transmitted by parental figures, but above all respect for the environment. In terms of Derryberry & Barger (2008) theoretical assumption on the moral development this task is located at a second level (the first level is focused on the satisfaction of personal needs and interests, the second one is oriented around norms and sanctions by authority, and the last level involving moral principles of justice and fairness).

Both contents (ethical and neutral) included indications, some necessary to solve the problem, others misleading, useless or contradictory. At the end of the passage there was a question that summarized the purpose of the reasoning. Then, questions were constructed (3 questions for each of the 5 phases of the paradigm) to which the subjects had to answer after reading the passages. These responses were used by the experimenter to verify the use of CT by the subjects in its various phases. The passages and questions were elaborated on the basis of the comparison between the different contributions on CT previously illustrated, taking into account the elements common to the different definitions.

The tracks presented are the following.

*Track n. 1* (neutral problem):

Marco is a child who attends primary school. As a child he had won a medal for swimming. Many of his friends already practice a sport, he would like to do it too, but he has to choose which one. Marco loves reading joke books and he hates video games. The town where he lives is not very big, but it has a sports field that offers the possibility of playing football. In fact, two of his classmates play in the local team: Simone, his best friend and Luca, for whom he does not feel much sympathy, and with whom he fights every day at school. Francesco, Marco’s best friend, would like him to go and play football and tells him that the coach is really good. Marco’s mother, in the pink and orange mailbox, found a flyer advertising swimming lessons in the nearby village pool. Marco learned to swim in the sea last summer to please his uncle who is a professional and kept insisting. Rather than swimming in green water, he preferred the diving competition which, however, is not included in the course in the pool.

At school during physical education lessons, Marco really likes to play basketball because, being a little taller than his other teammates, he is better at scoring baskets. The teacher always compliments him!

In the middle school gym, there are basketball training sessions and on Saturday there is a match with the other teams. Riccardo plays tennis very well, but he tells Marco that he is not interested in sports, it is a waste of time, he prefers to go home from school and play on the PlayStation.

If you were Marco, what would you choose?

*Track n. 2* (problem with ethical implications):

Alice and Matteo are two children who attend fourth grade, they get along very well and often find themselves at Alice’s home to watch TV. Alice is a little girl with a sweet tooth, especially for fruit candies, she never stops eating them even if they always give her a stomachache! Matteo is also a glutton, but above all for spaghetti with tomato sauce, he always manages to take it with him! One Sunday in the summer, Alice and Matteo, with their families, go to the park near their home, taking the Pokémon stickers with them. The Park is beautiful, they had just cut the grass and there are many colorful flowers. In their backpacks, of course, there is a secret supply of all-fruit candies for Alice and food of every taste for Matteo. Alice and Matteo argue constantly, but every now and then they climb together to the top of a tree as high as a skyscraper to refuel with sweets and chocolates, but then the wrappers are always around; where can they throw them? The litter-baskets are there, but never very close to where they are. Alice throws them among the leaves, Matteo instead puts them in his pocket. Soon it was time for a snack, two good chocolate tarts and a peach fruit juice, then they decide to unwrap the packs of football cards they have brought with them and exchange them with each other. They eat, play and the cards remain on the grass. Their mothers say to go and throw everything away because the place around them is really full of dirt, but they always postponed saying: “Yes Mom, we’ll go, the trash can is over there!” It’s time to go home, after the repeated requests of their parents, Alice and Matteo, backpacks on their shoulders, finally decide to go to the litter-basket to throw everything away, but they meet their friend Marco, so they leave everything on the grass and do some kicking with the ball.

Soon after, the voices of the parents are heard calling them to go home.

What would you do?

Questions related to the CT test:

Clarification: What kind of problem is this? What is the central point of the problem? Can you think of a similar problem to this?

Analysis: What are the parts of this problem? What are the links among the information you have read? What are the links with the final question?

Evaluation: What information can you believe and what information is not credible? Is there any unhelpful information that is not needed to solve the problem? Does all the information contained in the problem agree or contradict itself?

Inference: What result have you achieved? What information do you base your result on? Could you think of something different as a solution?

Self-monitoring: Was it difficult to arrive at the solution? Did you check if you thought correctly? Have you found something that does not convince you?

Both texts were counterbalanced and presented in random order. For each correct reply, 1 point was assigned. It results that the maximum for each phase of CT is 3. The Table 2 shows an example of scoring.

### 2.4. Procedure

First, the children’s parents were asked for written informed consent. All subjects gave their informed consent for inclusion before they participated in the study. The study was conducted in accordance with the Declaration of Helsinki. All the subjects of the school, 932 children, underwent the psychometric test of Raven Progressive Matrices [60]. The children performed the test in the presence of both the experimenter and the teachers. A maximum time of 30 min was allowed for completion of the test.

Following administration of the Raven Progressive Matrices test, the subjects belonging to the three groups, with the exception of the 18 children in the sample attending the first classes, were gathered in small groups in a classroom and given the CT test.

The experimenter, after having distributed the material containing the two passages and the related questions to each child, gave them the assignment, asking the children to read the first passage carefully and, at the end of the reading, to answer the related questions. The same procedure was repeated with the second piece. The maximum time allowed for the test was 60 min. It was decided to give the test verbally (by reading it) to the 18 children of the sample attending the first classes, in order to overcome the difficulties that the children would have encountered in reading the passages, since this is a capacity that is not yet sufficiently automated. There was a total of 182 children from the first class. Six of them were gifted children and we equaled them with 6 non-gifted chronologically age matched and 6 non-gifted mental age matched children. The experimenter personally administered the test to the children, reading them the passages, and subsequently transcribing the relative responses. No statistical differences were found on CT performance between children who read on their own compared with children to whom the experimenter read the instructions.

### 2.5. Statistical Analysis

Data were analyzed using SPSS Version 24.0 for Mac. Measurement parameters were the mean of correct responses (CR) for each phase of CT for each participant, for each experimental condition. Data were analyzed using the SPSS 24 statistical program, with a design of multivariate mixed model analysis of variance of repeated measurements: 3 (gifted children, chronologically age matched group, mental age matched group) × 2 (neutral content vs ethical content) × 5 (critical thinking phases: clarification, analysis, evaluation, inference, self-monitoring); for specific comparisons t-tests were used.

Descriptive statistics of the dependent variables were tabulated and examined. Alpha level was set to 0.05 for all statistical tests. In the case of significant effects, the effect size of the test was reported. For ANOVA, partial eta-squared ηp2 was used, and for the t-test, Cohen’s d Effect Size was used. The Greenhouse–Geisser adjustment for non-sphericity was applied to probability values for repeated measures.

## 3. Results

Table 3 presents data on means (M) and standard deviations (SD).

Groups showed significant effects: [F (2, 89) = 5.02, *p* < 0.01, ηp2 = 0.062]. This indicates that the performance in CT of the children shows significant differences. Gifted children differ from the group of chronologically age matched children, t (88) = 4.28, *p* < 0.01, d = 0.93, but not in respect to age matched children, t (88) = 2.11, *p* < 0.11, d = 0.91.

Furthermore, Phases shows significant effect [F (4, 166) = 5.02, *p* < 0.01, ηp2 = 0.071], this means that some phases of CT are higher than others. The interaction Groups X Phases showed a significant effect: [F (8, 344) = 4.11, *p* < 0.01, ηp2 = 0.075]. This indicates that the performance in CT of the gifted children is higher than chronologically age matched children only in some phases: clarification and evaluation. In these phases, gifted children have a higher-level performance than chronologically age matched children. This is because gifted children are more able than nongifted children to analyze detailed information and clarify the setting and are also very able in the evaluation phase that is the most difficult phase of CT as it consists of accepting the value of the sources to verify the reliability and credibility of the information. Figure 1 and Figure 2 show the results of the three groups. Furthermore, no significant differences were found between the ethical and the neutral conditions. It may be that the general high ability to perform complex reasoning task is not influenced by the context in which it is applied. These results indicate that it is mainly mental age and not IQ that has a fundamental influence on the development of the critical thinking process.

## 4. Conclusions

The aim of this research was to evaluate the ability to use critical thinking in mentally intellectually gifted subjects and to compare their performance to intellectually nongifted subjects paired by chronological age and nongifted subjects paired by mental age.

The results show that gifted children provide better performance than nongifted ones, in agreement with the main hypothesis that gifted children have higher complex cognitive processes than nongifted children [32,33]. In fact, giftedness predicts CT skills and also the ability to solve problems [10,11]. Furthermore, these findings are also explained by the fact that gifted students use a wider variety of strategies when solving problems than their peers [36,37] and they are generally able to keep their attention on a problem or task in ways that their non-gifted peers cannot [41,42,43,44,45,46]. These specific characteristics of intellectually gifted children could make them more critical thinkers than their average ability peers, and therefore perform better in critical thinking their average ability peer.

Specifically, analysis of the individual phases related to the CT process shows that the clarification and evaluation phase (first and third phases) present significant differences between the various groups of subjects both in neutral content and ethical content. This analysis shows that intellectually gifted subjects understand more the different parts of the problems and identifying whether these parts can be considered sources of useful and reliable information. The evaluation phase is deemed the most difficult phase and requires higher intellectual skills, which are lacking in the average child.

The existence of such significant differences between gifted and both nongifted children are not confirmed by a more general comparison between the three groups of subjects in relation to the two types of CT (neutral and ethical).

The initial purpose, to verify that intellectually gifted children differ from both normally intellectual children in the use of CT, is confirmed in the five phases related to CT. In particular, the ability to evaluate sources is the ability in which gifted subjects are most efficient, among the different phases of CT identified by Paul and Scriven [63] and Facione [13,62]. Thus, gifted subjects are more adept at understanding problems in different settings and are more capable to understand both ethical and neutral implications. Gifted individuals appear to be more adept at identifying the reliability and credibility of the sources of information and any contradictions or conflicts between them. This ability is considered by Jones [61] and Ennis [1] as the fundamental skill in the process of critical thinking.

These findings are in line with literature indicating that gifted children possess higher levels of cognitive processes but, at the same time, do not add anything to the literature on moral processes [36,37]. In this case, the greater ability in clarifying and evaluating the sources of information phases gives them the possibility to succeed in critical thinking abilities. As noted above, this advantage in critical thinking can be viewed as more efficient ability to capture the right sense of critical thinking.

These findings make a new contribution to the previous literature [16,18,20,27,28,29,63] on the study of CT in intellectually gifted children, because existing research has focused on studying the development of CT in intellectually average children, but not on intellectually gifted children [47,51,62,63]. Specifically, the innovative contribution to the literature was to investigate possible differences in the CT of intellectually gifted versus intellectually nongifted children by exploring the precise phases of CT (clarification, analysis, evaluation, inference, self-monitoring) that differentiates gifted from intellectually nongifted children [62,63].

With regards to the ethical component, as previously mentioned, it emerged that the capacity for critical thinking was the same in ethical and neutral contexts, so this result is in line with the study of Beißert & Hasselhorn (2016) in which emerged that moral developmental status seems to be independent from children’s general intelligence [58]. This could explain the result obtained, but this aspect of the research does not add anything to the existing literature on moral processes [48,49,50,52,55,56,57].

Although gifted children may have some strengths, many studies have failed to find differences between this population and their typically developing peers on many other cognitive measures. Further studies are needed to better understand this heterogeneity and to explore the moral questions related to complex reasoning better.

Despite the results that emerged from this research, some limitations should be highlighted. The first is the small sample size so therefore it may be useful to replicate the experiment with a larger sample.

The second is that the present study, as mentioned above, does not add anything to the literature on moral processes. We think that more specifically addressed on the ethical tasks can be useful.

In the context of child development and education, considering that CT is a mental process useful to solve problems [3,4,5,10], it would be hopefully to use specific strategies in schools to promote the development of this mental processes in children to become critical thinkers in adulthood. In fact, the skill of CT can help children and adults to overcome prejudices and errors of thinking; for this reason, it can be important to expose children to it since in early childhood. Instead, with reference to the moral processes, it could be useful to create training to increase ethical thinking because it is not necessary to be gifted to be moral, and the gifted are capable of incredibly destructive and immoral behavior. In fact, cognitive and intellectual development have been recognized as necessary but not sufficient conditions for the growth of moral judgment [64,65,66,67]. For example, it could be useful to do in primary school metacognitive trainings reinforcing different abilities involved in moral judgments (mentalizing abilities, executive abilities) [68].

## Figures and Tables

**Figure 1 children-10-00074-f001:**
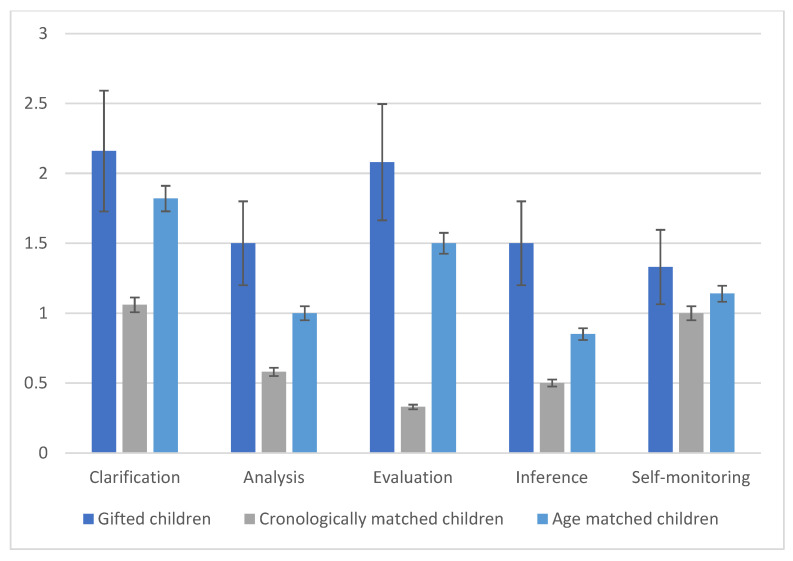
Means of the correct performance on the five phases of neutral critical thinking.

**Figure 2 children-10-00074-f002:**
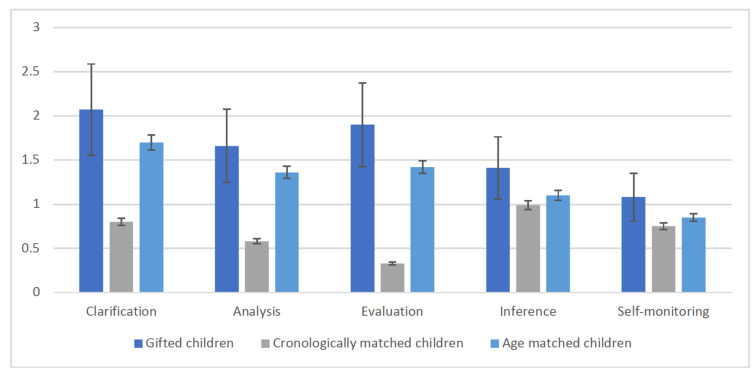
Means of the correct performance on the five phases of ethical critical thinking.

**Table 1 children-10-00074-t001:** Socio-demographic Characteristics of Participants at Baseline.

	Gifted Children	Chronologically Matched Children	Mental Age Matched Children
Female	17	17	14
Male	15	15	13
Age (M(SD))	9.1 (±1.02)	9.2 (±0.98)	11.8 (±1.02)
Age range IQ (M(SD))	(7.2–11.7) 137.6 (±6.2)	(7.2–11.7) 107.3 (±8.10)	(10.5–11.9) 105.4 (±7.9)

**Table 2 children-10-00074-t002:** Example of scoring for each question of the neutral CT.

	First Question	Second Question	Third Question	Replies	Scores
**Clarification**	What kind of problem is this?	What is the central point of the problem?	Can you think of a similar problem to this?	1st It is a choice	1
				2nd The choice of the best sport for Marco	1
				3rd To choose future school	1
**Analysis**	What are the parts of this problem?	What are the links among the informa-tion you have read?	What are the links with the final question?	1st Marco has to choose sport, he attends primary school, he has friends and family that may influence him	1
				2nd Time, places where to do sports and friends	1
				3rd How these three things affect the choice	1
**Evaluation**	What information can you believe and what information is not credible?	Is there any unhelpful information that is not needed to solve the problem?	Does all the information contained in the problem agree or contradict itself?	1st I can’t believe Marco’s best friend, because there are two different names for his best friend	1
				2nd it is not relevant the color of the mailbox: pink and orange	1
				3rd There are some that agree … and some that contradict itself…	1
**Inference**	What result have you achieved?	What information do you base your result on?	Could you think of something different as a solution?	1st May be basketball	1
				2nd the fact that he is taller than his teammates and he reach high scores	1
				3rd Yes. I can think about Swimming but there are some contraddiction	1
**Self-monitoring**	Was it difficult to arrive at the solution?	Did you check if you thought correctly?	Have you found something that does not convince you?	1st Yes, a little bit, maybe too many information	1
				2nd Maybe I had a mistake in remembering all the names, I can check again if I have time	1
				3rd Yes. For example, the swimming question	1

**Table 3 children-10-00074-t003:** Means (and standard deviations) of correct performance on the five phases of critical thinking.

	Gifted Children	Chronologically Matched Children	Age Matched Children
**Neutral**			
Clarification	2.16(±1.19)	1.06(±1.23)	1.82(±0.97)
Analysis	1.60(±1.12)	0.58(±1.60)	1.00(±0.53)
Evaluation	2.08(±0.90)	0.33(±0.49)	1.78(±0.75)
Inference	1.60(±1.11)	0.50(±1.00)	0.85(±1.21)
Self-monitoring	1.33(±1.02)	1.00(±0.73)	1.14(±0.89)
**Ethical**			
Clarification	2.07(±0.99)	0.80(±0.79)	1.70(±0.69)
Analysis	1.66(±0.98)	0.52(±0.62)	1.57(±0.97)
Evaluation	1.80(±0.95)	0.25(±0.45)	1.42(±0.53)
Inference	1.41(±1.24)	1.00(±1.12)	0.57(±0.78)
Self-monitoring	1.08(±0.79)	0.75(±0.75)	0.85(±0.69)

## Data Availability

Not applicable.

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
