# Peer review of "Critical Thinking in Ethical and Neutral Settings in Gifted Children and Non-Gifted Children"

_children, 2022, doi:10.3390/children10010074_

Round 1
Reviewer 1 Report
The text is well written, the procedure is clearly described and logically ordered.
The introduction and conclusion is connected to the research conducted in the field.
I suggest authors consider the possibility to mention how the sample was reduced, what were the criteria for the selection (random / non-random selection with the 3 groups).
Author Response
Reviewer 1 Comments
- I suggest authors consider the possibility to mention how the sample was reduced, what were the criteria for the selection (random / non-random selection with the 3 groups).
Response 1 for Point 1
We added more clarifications to the phases of reduction of the sample:
The initial sample of subjects was made up of 932 children attending a primary school, 530 females and 402 males, from three primary schools in Milan, of medium socio-economic status, attending from the first to the fifth grade. The Raven Intelligence Measurement Test - Progressive Matrices [61] was administered to all 932 children, and on the basis of the IQ scores obtained (130), the experimental group was formed with 32 children with a mean IQ = 137.6 (± 6.2). Following this first screening for giftedness, we obtained a non-gifted sample of 900 children. From this remaining group we extrapolated firstly 32 normally developing children matched for chronological age and gender; each child of this second group was matched one-by-one with each gifted child based on the same chronological age and the same gender; from the same group of 900 non-gifted children we extrapolated secondly 27 normally developing children matched for mental age and gender; each child of this third group was matched one-by-one with each gifted child based on the same mental age and the same gender (Table 1). The first chronologically age matched control group (mean CA = 9.12 (± 0.98)), consisted of 32 intellectually nongifted children, with the same chronological age and the same gender of gifted children (expressed in years and months). Then, the second mental age matched control group consisted of 27 intellectually non-gifted children, matched by gender and mental age compared to the gifted group, but with a higher chronological age (mean CA= 11.8 (± 1.02)).
Reviewer 2 Comments
Point 1
- The authors would perhaps benefit from developing the section on moral development. Indeed, many moral development tests rely on verbal abilities (Karnes & Brown, 1980; Sanders et al., 1995). In addition, authors could benefit from specifying the level (Derryberry & Barger, 2008) in which the task they are using is located (a first level focused on the satisfaction of personal needs and interests, a second level oriented around norms and sanctions by authority, and the last level involving moral principles of justice and fairness.
Response 1 for Point 1
Thank you very much. We enriched our literature with studies that go more in depth on moral development such as:
- Beißert, H. M., & Hasselhorn, M. (2016). Individual differences in moral development: Does intelligence really affect children’s moral reasoning and moral emotions?. Frontiers in psychology, 7, 1961.https://doi.org/10.3389/fpsyg.2016.01961.
Derryberry, W. P., & Barger, B. (2008). Do contributors to intellect explain the moral judgment abilities of gifted youth?. Gifted Child Quarterly, 52(4), 340-352. https://doi.org/10.1177/0016986208321806.
Karnes, F. A., & Brown, K. E. (1980). Moral development and the gifted: An initial investigation. Roeper Review, 3(4), 8-10. https://doi.org/10.1080/02783198109552540.
Sanders, C. E., Lubinski, D., & Benbow, C. P. (1995). Does the Defining Issues Test measure psychological phenomena dis- tinct from verbal ability? Journal of Personality and Social Psychology, 69, 498-504. https://doi.org/10.1037/0022-3514.69.3.498.
Point 2
- I would have appreciated it if there could have been the minimum and maximum age of children in this school (and for each group). Also, since the procedure is slightly different between children in 1st grade, how many are there in that case? Could this not have an impact on the results?
Response 2 for Point 2
Thank you
We added the range of age for each group in the table 1.
There was a total of 182 children from the first class. Six of them were gifted children, 6 were non- gifted chronologically age matched and 6 non-gifted mental age matched they were in total 18 children. The experimenter gave the test verbally (by reading the texts) to the 18 children of the sample attending the first classes, in order to overcome the difficulties that the children would have encountered in reading the passages, since this is a capacity that is not yet sufficiently automated in non-gifted children. No statistical differences were found on CT between children who read on their own compared with children to whom the experimenter read the instructions.
We added in the text the exact number of first-class children and we also added the absence of statistical difference between the two modality of presentation.
Point 3
- In terms of procedure, the authors do not specify whether there is a counterbalancing of the texts and there is a lack of details on how the evaluation of the children's responses was carried out.
Response 3 for Point 3
We added the counterbalancing of the texts in the text.
For each correct reply, 1 point was assigned. It results that the maximum score for each phase of CT is 3. Moreover, we added the table 2 in which we show an example of scoring.
Point 4
- On the results, the text would gain in precision by providing the effect sizes as well as the statistics used for specific comparisons. Finally, the results figures could be improved by adding standard deviations or measurement error bars.
Response 4 for Point 4
Thank you! We completely forgot to add the statistical analysis section. We added this section and provided the effect sizes and the statistics used for specific comparisons.
Thank you also for the figures, we added error bars.
- The discussion would benefit from analyzing the results in relation to more references and from giving indications on the theoretical contribution of these results as well as on their contribution to the care of children with high intellectual potential.
REVIEWER 3
Point 1
- it is necessary to more clearly and unambiguously formulate the purpose of the study.
Response 1 for Point 1
We better formulated the purpose of the study as follows:
In particular the research objectives were:
- In relation to the results of previous literature [38,44,45,46] our aim was to investigate whether gifted children show a higher level of performance in CT abilities than non-gifted children;
- To examine whether in some phases of CT (clarification, analysis, evaluation, inference, self-monitoring), and possibly in which ones, the differences described above are particularly significant.
- To understand if significant differences emerge in the ability to use CT, between gifted and nongifted children in two different types of settings one ethically neutral and the other with ethical implications [64].
Point 2
- It is also not clear why the first task of the study contains links to publications of other authors: “To study if gifted children show a higher level of performance in CT abilities than non-gifted children [38,44,45,46];”
Response 2 for Point 2
We explained that the first hypothesis was related to that precise literature cited [38,44,45,46].
Point 3
- However, it is necessary to present the limitations as well as practical applications of this study in the Conclusions.
Response 3 for Point 3
We added the limitations.
Point 4
- In addition, I recommend specifying the title of the article:
“Features of critical thinking in ethical and neutral settings in gifted and тот-gifted children”
Response 4 for Point 4
Thank you, we accepted your suggestion.
REVIEWER 4
- It would be informative for the readers to comment also the effect of social environmental factors (e.g. characteristics of the family – SES, family structure, ethnic/migrant background, parental involvement in child education and care, etc.) on child development, resp. giftedness and critical thinking. In this respect, you may also add a paragraph with a description of the limitations of the study and potential future directions of this research. Apart from the contribution to the existing research, you can discuss the implications of the study findings for the practice in the sphere of child development and education. Overall, the paper is very well written but additional references to the influence of social environmental factors on the psychological development of children in the studied domains would further strengthen and better delineate the contribution of the study to the respective research field.
Response for the points of reviewer 4
We added some social characteristics of the sample. We also added the limitations of the study.
The implications of the study findings for the practice in the sphere of child development and education were added in the discussion section.

Reviewer 2 Report
The article "critical thinking in ethical and neutral settings in gifted children" is very interesting because it is the first, to my knowledge, to compare the performance of CT for gifted children vs. nongifted children (with 2 control groups).
The review of the literature is relevant. The authors would perhaps benefit from developing the section on moral development. Indeed, many moral development tests rely on verbal abilities (Karnes & Brown, 1980; Sanders et al., 1995). In addition, authors could benefit from specifying the level (Derryberry & Barger, 2008) in which the task they are using is located (a first level focused on the satisfaction of personal needs and interests, a second level oriented around norms and sanctions by authority, and the last level involving moral principles of justice and fairness. Moreover, Beißert & Hasselhorn, (2016) recently found an absence of correlation between moral development and nonverbal intelligence.
I congratulate the authors for the procedure used to create the three groups. Indeed, the distribution of HPI is slightly higher than expected (3.43% instead of 2.5%) but above all, it allows to highlight a balance between boys and girls. However, I would have appreciated it if there could have been the minimum and maximum age of children in this school (and for each group). Also, since the procedure is slightly different between children in 1st grade, how many are there in that case? Could this not have an impact on the results?
In terms of procedure, the authors do not specify whether there is a counterbalancing of the texts and there is a lack of details on how the evaluation of the children's responses was carried out.
On the results, the text would gain in precision by providing the effect sizes as well as the statistics used for specific comparisons. Finally, the results figures could be improved by adding standard deviations or measurement error bars.
The discussion would benefit from analyzing the results in relation to more references and from giving indications on the theoretical contribution of these results as well as on their contribution to the care of children with high intellectual potential.
Beißert, H. M., & Hasselhorn, M. (2016). Individual differences in moral development: Does intelligence really affect children’s moral reasoning and moral emotions?. Frontiers in psychology, 7, 1961.https://doi.org/10.3389/fpsyg.2016.01961.
Derryberry, W. P., & Barger, B. (2008). Do contributors to intellect explain the moral judgment abilities of gifted youth?. Gifted Child Quarterly, 52(4), 340-352. https://doi.org/10.1177/0016986208321806.
Karnes, F. A., & Brown, K. E. (1980). Moral development and the gifted: An initial investigation. Roeper Review, 3(4), 8-10. https://doi.org/10.1080/02783198109552540.
Sanders, C. E., Lubinski, D., & Benbow, C. P. (1995). Does the Defining Issues Test measure psychological phenomena dis- tinct from verbal ability? Journal of Personality and Social Psychology, 69, 498-504. https://doi.org/10.1037/0022-3514.69.3.498.
Author Response

(The authors gave the same response as above.)

Reviewer 3 Report
This study is devoted to a very relevant topic and has an interesting experimental design.
In the Introduction, the authors presented a comprehensive review of the literature that sufficient to formulate the problem of the experiment. However, at the end of this section, it is necessary to more clearly and unambiguously formulate the purpose of the study. In the presented version, we see several different formulations of the aims of the study:
- “In particular, we aim to measure the performances of gifted children in the five phases of CT: clarification, ability to focus on a problem and give it meaning; analysis, ability to identify the relationship between the parts of a problem and to distinguish what is relevant from what is not; evaluation, ascertain the value of the sources to verify reliability, agreement, and credibility of information; inference, come to a coherent and reasonable conclusion based on the information analyzed; self-monitoring, knowing how to mon-itor cognitive procedures throughout the process, starting from self-observation up to self-correction.”
- “The aim of this research was identifying differences between intellectually gifted children and intellectually non-gifted children in the use of CT.”
- “The general aim of this research was to verify the existence of any differences in the ability to use CT by intellectually gifted children com-pared to intellectually average children.”
It is also not clear why the first task of the study contains links to publications of other authors: “To study if gifted children show a higher level of performance in CT abilities than non-gifted children [38,44,45,46];”
The content of other sections of the article complies with the accepted rules.
The results are clearly presented in tables and figures. However, it is necessary to present the limitations as well as practical applications of this study in the Conclusions.
In addition, I recommend specifying the title of the article:
“Features of critical thinking in ethical and neutral settings in gifted and тот-gifted children”
Author Response

(The authors gave the same response as above.)

Reviewer 4 Report
The paper presents findings from a study that aims to evaluate the ability to use critical thinking in mentally intellectually gifted and nongifted children, paired by chronological age (gifted subjects) and by mental age (nongifted subjects). The experimental study has a split-split-plot design - one between subjects and two within subjects. The study is very well designed with a clear description of the empirical setting, the sampling and matching procedures and the statistical methods used in the analysis. The findings are clearly presented and framed in the context of existing research.
I have the following suggestions to the authors. The study includes children from middle-class families divided and matched in groups based on their giftedness and other characteristics. It would be informative for the readers to comment also the effect of social environmental factors (e.g. characteristics of the family – SES, family structure, ethnic/migrant background, parental involvement in child education and care, etc.) on child development, resp. giftedness and critical thinking. In this respect, you may also add a paragraph with a description of the limitations of the study and potential future directions of this research. Apart from the contribution to the existing research, you can discuss the implications of the study findings for the practice in the sphere of child development and education. Overall, the paper is very well written but additional references to the influence of social environmental factors on the psychological development of children in the studied domains would further strengthen and better delineate the contribution of the study to the respective research field.
Author Response

(The authors gave the same response as above.)
